# The Degradation of Intramuscular Connective Tissue In Vitro with Purified Cathepsin L from Bovine Pancreas

**DOI:** 10.3390/foods12183517

**Published:** 2023-09-21

**Authors:** Yingbo Peng, Wanhong He, Shuang Teng, Muneer Ahmed Jamali

**Affiliations:** 1College of Engineering, Nanjing Agricultural University, Nanjing 210095, China; 2College of Food Science and Technology, Nanjing Agricultural University, Nanjing 210095, China; 3National Center of Meat Quality and Safety Control, Nanjing Agricultural University, Nanjing 210095, China; 4Department of Animal Products Technology, Sindh Agriculture University, Tandojam 70060, Pakistan

**Keywords:** intramuscular connective tissue, cathepsin L, collagen, decorin, degradation

## Abstract

To investigate the possible degradation of the intramuscular connective tissue (IMCT) with cathepsin L, isolated IMCTs were incubated with purified cathepsin L in vitro. Here, we prepared purified cathepsin L from bovine pancreas by using DEAE Sephacel, Sephacryl S-100 HR, SP Sepharose FF, and con A-Sepharose affinity chromatography in sequence. An SDS-PAGE analysis of CNBr-digested peptides showed that the degradation of collagen in IMCT could take place on terminal non-helical peptides rather than the triple helix region. Decorin (DCN) was clearly degraded at a pH of 5.0. The T_P_ and T_O_ of intramuscular connective tissue decreased to 41.41 °C and 43.79 °C, respectively. In the cathepsin L treatment of pH 5.0, the decreases in the T_P_ and T_O_ of IMCT were more sensitive than they were at pH 5.5~6.5.

## 1. Introduction

Meat tenderness is a critical factor for consumer acceptance, which determines repeat purchasing, satisfaction, and willingness to pay premium prices [1]. The IMCT, which is mainly composed of fibrillar collagens embedded in a matrix of proteoglycans (PGs) [2], provides biomechanical strength and makes a key contribution to meat toughness [3]. It is distributed in a network structure between muscle fibers and muscle bundles, playing an important role in maintaining the integrity of muscle structure, and is called “background toughness”.

A growing number of researchers have shown that collagen concentration could not explain shear force variation. Christensen et al. [4] also reported no correlations between collagen characteristics and Warner-Bratzler shear force. Hopkins et al. [5] found no correlation between the collagen concentration and the shear force in semitendinosus in lamb. Thus, the thermal transition of IMCT was considered. Aktaş and Kaya [6] observed a decline in the T_O_ and T_P_ of IMCT with the decrease of pH. In addition, Latorre et al. [7] observed thermal variability (activation energy) of IMCT-perivascular collagen in different muscles. The thermal properties determined for IMCT were primarily attributed to mature cross-links and extracellular modifications [2,6,8,9].

Cathepsins B, H, L, and X are a group of enzymes categorized into cysteine, and they are present in the lysosomes of animal tissues. The role and activity of cathepsins during post-mortem storage or low temperature meat processing is unclear [10]. Nevertheless, it is clear that cathepsins are hydrolyzed on different proteins, such as myofibrillar, myosin, actin [11], and even collagen proteins [12,13]. There is growing evidence that cathepsin L also participates in the changes of collagen. Cathepsin L cleaves degraded/unfolded collagens like gelatin. It is known that cathepsin L can cleave collagen and myofibrillar proteins and that it must be released from lysosomes to contribute to meat tenderization [14]. Beltrán et al. [15] found that cathepsin L could degrade proteoglycan in collagen or directly hydrolysis collagen. Nishimura et al. [16] showed that PGs were degraded in the post-mortem aging of beef. PG degradation appeared to be the primary factor in the IMCT weakening and the partial tenderization of beef. It is little known how cathepsin L hydrolyzes connective tissue and what changes occur in cross-linking, collagen, and proteoglycans during the degradation. To our knowledge, there have been no studies on the hydrolysis of decorin (the major type of PG in striated muscle) by cathepsin L. Differential scanning calorimetry (DSC) was used to explore the thermal properties of IMCT in this study. A decrease in the transition temperature of IMCT was observed when it was incubated with cathepsin [16] or weak organic acids or NaCl [17]. The specific mechanism underlying the decrease in the thermal transition temperature of IMCT in the presence of cathepsin L has not been well elucidated.

Physiological maturity and intramuscular fat deposition can affect beef tenderness. On the one hand, as physiological maturity increases, shear force increases and tenderness deteriorates; on the other hand, as intramuscular fat becomes deposited, shear force decreases and tenderness improves. For the explanation of this phenomenon, there is currently only the theory that overdeveloped adipocytes “destroy” the structure of connective tissue. Considering that fat particles grow and develop within adipocytes, and adipocytes have a certain fluidity, mechanical destruction is not the fundamental reason for tenderness change. At the level of connective tissue, the content of collagen is not significantly related to the shear force value of meat; thus, the goals of this work are as follows: (1) to demonstrate the degradation of collagen and decorin in IMCT that was incubated with purified cathepsin L from bovine pancreas in vitro; (2) to observe the changes in the thermal transition of IMCT in the presence of cathepsin L. The findings of this study would help to explain the possible mechanism for IMCT weakening after conditioning.

## 2. Materials and Methods

### 2.1. IMCT Preparation

*Longissimus lumborum* (*LL*) were humanely harvested from the carcasses of approximately 2.5 years old beef cattle at post-mortem 36 h from Qinbao Animal Husbandry Development Co., Ltd. (Shaanxi, China) and stored at −20 °C until sampling. The samples were thawed at 4 °C. In this study, the medial central portion of the muscle was used for analysis. IMCT was prepared and purified by the method of Loveday et al. [18]. An amount of 400 g of chopped fresh muscle was homogenized with 20 mM phosphate buffer (pH 6.1, 1 mM phenylmethanesulfonyl fluoride (PMSF)) and 0.05 M NaCl (buffer A) in a Waring blender (8010ES, Waring Commercial, Stamford, CT, USA). Then, the dispersion was filtered through a size 25 mesh. The connective tissue fibers were stirred into 500 mL of a 0.05 M NaCl solution containing 1 mM PMSF overnight. After centrifugation (3000× *g*, 20 min, 4 °C) (Centrisart^®^ D-16C, Sartorius, Göttingen, Germany), the supernatant was discarded. The precipitate was homogenized again, and then, the resulting precipitate was stirred for 12 h and centrifuged (3000× *g*, 20 min, 4 °C). After it was washed, the purified connective tissue was lyophilized and stored at −80 °C until analysis.

### 2.2. The Preparation of Purified Cathepsin L

The purification procedures were executed according to Tang et al. [19], Li et al. [20] and Gillies and Lieber [21], with modifications. In brief, the bovine pancreas was minced, homogenized in a 50 mM sodium acetate buffer (pH = 5.0), and centrifuged (4000× *g*, 15 min, 4 °C) twice. The supernatants were combined, and procedures for acidification and ammonium sulfate precipitation were followed. After being centrifuged, dialyzed, and concentrated, the concentrated samples were sequentially passed through chromatography by DEAE Sephacel anion exchange, Sephacryl S-100 HR size exclusion, and SP Sepharose FF cation exchange to remove large molecular weight heteroproteins, followed by con A-Sepharose affinity chromatography to obtain cathepsin L [20,21,22]. One unit of enzyme activity was defined as the amount of activity that released 1 nmol of 4-Methyl-7-aminocoumarin (AMC) per min under assay conditions. The active eluate of cathepsin L was collected, concentrated, and stored at −80 °C for subsequent identification by SDS-PAGE. Optimum pH was determined according to Wang et al. [22]. Briefly, the buffer pH was assayed in the range of 4.0–7.5, Z-Phe-Arg-MCA was used as substrates for cathepsin L, and optimum pH was determined by the activity of cathepsin L.

### 2.3. The Incubation of IMCT with Purified Cathepsin L, CNBr Digestion, and SDS-PAGE

Approximately 5 mg of lyophilized IMCT (dry wt.) was allowed to rehydrate overnight in water before incubation. This material was pressed dry and incubated with purified cathepsin L (50 units) in different citrate buffers (20 mM, 10 mM cysteine, and 20 mM CaCl_2_) with different pH values (5.0, 5.5, 6.0, and 6.5). All of the marinating solutions were adjusted to approximately 1 mL. These pH value settings were employed to simulate pH changes that occur during the post-mortem aging of beef and thus were not necessarily optimum values for extracted, purified cathepsin L. The same procedures were performed without enzymes, and the resulting samples were used as controls. After 12 h of incubation at 30 °C, the treated IMCTs were thoroughly cleaned in Millipore water for 20 s (twice). To explore possible variations over time, the incubation procedures were also executed as follows: pH 5.0 and 5.5 for 12, 24, and 36 h.

After incubation, all of the treated IMCTs were subjected to a CNBr digestion as described by López et al. [23], Scarr [24], Gillies and Lieber [21], and Vidal [25]. Each treated IMCT sample was filled with a solution to a volume of 1.2 mL (70% formic acid) and 0.3 mL (50 mg/mL CNBr) to produce a final 10 mg/mL CNBr. Nitrogen gas was then bubbled in, and the tubes were sealed. The reaction was allowed to proceed overnight at 25 °C. The digests were centrifuged, and the supernatants were removed and lyophilized. The resulting peptides were then separated by SDS-PAGE by following the method described by Vidal [25]. The SDS-PAGE was performed using a 12.5% acrylamide separating gel and a 4% acrylamide stacking gel. After electrophoretic development, gels were stained with Coomassie Brilliant Blue R-250 and destained overnight with a solution (10% (*v*/*v*) methanol and 10% (*v*/*v*) acetic acid). The electrophoresis film was scanned by the gel system and analyzed by Quantity One (Bio-Rad, Hercules, CA, USA).

### 2.4. A Western Blot Analysis of the Degradation of Decorin (DCN) Incubated with Cathepsin L In Vitro

First, total protein was extracted with lysis buffer (50 mM Tris, pH 7.5; 250 mM NaCl; 5 mM EDTA; and 1% Nonidet P-40) according to the method described by Suzuki et al. [26]. A protein assay was used to determine the protein concentrations according to the methods described by text kit (Bio-Rad) and to ensure equal loading. After extraction, the whole protein samples were extensively dialyzed against a series of sodium acetate buffers (20 mM, 5 mM EDTA, and 0.3 mM PMSF) with different pH values of 5.0, 5.5, 6.0, and 6.5 to provide incubation conditions that simulated the possible dynamic changes in pH that take place during the post-mortem aging of beef.

After the samples were dialyzed, their protein concentrations were adjusted to approximately 0.5 mg/mL. One milliliter of the total protein sample (10 mM cysteine and 20 mM CaCl_2_) was incubated with 25 units of purified cathepsin L at 30 °C for approximately 6 h. The same procedure was also used without enzymes for the control. To explore possible variations over time, incubation conditions were also applied as follows: pH 5.0 and 5.5 for 6, 8, 10, 12, and 24h.

Electrophoresis (a running gel of 10% acrylamide) was executed following the method described by Honardoust et al. [27] and Wang et al. [22]. After being fractionated by SDS-PAGE, the proteins were transferred onto polyvinylidene fluoride (PVDF) membranes (Bio-Rad Laboratories, Hercules, CA, USA) in a transfer buffer [26,28]. The membranes were blocked at 4 °C for 1 h with 5% BSA in tris-buffered saline and tween (TBST). After blocking, these membranes were exposed to the following main antibody overnight at 4 °C: a 1:1000 dilution of rabbit polyclonal anti-decorin antibody (sc-22753; Santa Cruz, CA, USA). Then, the membranes were washed (5 × 10 min) with TBST and incubated with a secondary antibody, a 1:10,000 dilution of peroxidase-conjugated goat anti-rabbit IgG (ZB-2301, ZSGB-BIO, Beijing, China). Finally, the blots were washed again, and the protein bands were detected with an ECL reagent and scanned with an Image Quant LAS4000 (GE Healthcare, Chicago, IL, USA).

### 2.5. Differential Scanning Calorimetry (DSC)

The IMCT treatments were the same as they were in the procedure described in Section 2.3. There was a small difference in the preparation of the control group; to simplify the experimental design for the control group, only 36 h treatments were applied without cathepsin L for DSC analysis. After the incubation, the samples were washed in water and blotted on filter paper. Approximately 10 mg was placed in an aluminum DSC sample pan. All of the DSC measurements were performed by the same person, who was advised to observe the sample surfaces when they had approximately similar moistures. The pans were sealed and heated from 30 to 90 °C at 5 °C/min by using Perkin-Elmer Diamond DSC (Perkin-Elmer, Waltham, MA, USA) based on the method described by Chang et al. [17] and Castrejón-Flores et al. [29]. An empty sample pan was used as a reference, and T_O_ and T_P_ values were determined. DSC analyses were performed with replicates for each treated group.

### 2.6. Statistical Analysis

Three repetitions of each experiment were carried out. The data were analyzed using one-way ANOVA and Duncan’s Multiple Range Test with SAS (SAS Institute Inc., Cary, NC, USA). Differences were considered significant at *p* < 0.05.

## 3. Results and Discussion

### 3.1. The Purification of Cathepsin L from Bovine Pancreas

The molecular mass of purified cathepsin L was approximately 30 kDa (Figure 1), which was nearly the same as that of cathepsin L from carp [30] and blue garden trevally [31]. Optimum pH was 5.5 for the hydrolysis of Z-Phe-Arg-MCA with purified cathepsin L.

### 3.2. CNBr-Digested Peptides from Cathepsin L-Treated IMCT

As shown in the reports provided by Sun et al. [32], and Kuivaniemi et al. [33], the peptides α(I)CB-8 and α(III)CB-5 in Figure 2 and Figure 3 were identified as type I and type III in IMCT. In comparison with the untreated IMCT samples (lane u) in both Figure 2 and Figure 3, there was some “fuzziness” in the low molecular weight bands. Nevertheless, the CNBr-digested peptides (above 10 kDa) of residual insoluble materials treated with cathepsin L at different pH values and times demonstrated almost the same digestion pattern. This finding likely indicated that the cleavage of collagen by purified cathepsin L could occur at the terminal non-helical peptides of type I collagen rather than the native triple helix region. The distribution of peptide segments above 10 kDa in the enzyme treated group is similar, regardless of whether it is low or high pH, indicating that the main chain structure of collagen did not show significant changes after treatment with pancreatic tissue protease L. This observation proves that the catalytic domain of collagenases has general proteolytic activities, but it alone cannot cleave triple helical collagens [34]. As for interstitial collagenase, it specifically hydrolyzed collagen and cleaved type I collagen within the native triple helix at a single site, generating ^3/4^N-terminal (TC^A^) and ^1/4^C-terminal (TC^B^) fragments. Despite the lack of response of the collagenase-treated group in this research, the pattern of CNBr-digested peptides from collagenases that were treated with insoluble IMCT would presumably be different from the cathepsin L-treated or untreated samples.

### 3.3. The Degradation of DCN

Protein polysaccharides can provide a certain hydration space around cells, and some bound water molecules can be bound around amorphous protein polysaccharides, which plays a certain role in the stability of connective tissue. Generally speaking, there can be chains around protein polysaccharides distributed around the core protein. These glycosaminoglycan chains are highly sulfated and carry negative charges. The less cross-linking, the lower the “maturity” of collagen; the more thermally unstable proteins, the more it is easy to dissolve during heating, and the stability of connective tissue becomes weaker. The degradation of proteoglycans, which act as bridges between the sarcolemma and endometrium, is an important reason for the weakening of connective tissue structure, thereby reducing shear force.

DCN is already known as the major proteoglycan in striated muscles [35]. It acts as a spacer during the lateral assembly of the collagen molecular structure and is important for retaining normal mechanical properties and tissue function [36]. In Figure 4A, a visible “fuzziness” of the bands below DCN after 8 h of incubation at pH values of 5.0 and 5.5 (lanes e and f) was observed, and there were inconspicuous changes at pH 6.0 and pH 6.5 (lanes g and f). This finding indicated that the DCN degradations by purified cathepsin L were sensitive to pH values of 5.0 and 5.5. In Figure 4B,C, the degradation of DCN was intense after 24 h of incubation. Bands (<43 kDa) were observed in lanes g, h, i, and j, indicating that a specific site in DCN was cleaved by cathepsin L. Figure 4B,C showed that an obvious band (<43 kDa) appeared earlier at pH 5.0 than at pH 5.5, indicating that DCN was sensitive to hydrolysis at a low pH. According to Robinson et al. [37], PGs are associated with collagen fibrils in the endomysium and the perimysium. In this work, the extracellular modification of PGs to collagen was possibly weakened by the degradation of DCN, which might have induced the destruction of the IMCT and decreased the stability of the collagen. These results implied the weakening of IMCT during postmortem conditioning with cathepsins.

### 3.4. The Analysis of IMCT by DSC

To refers to the initial denaturation temperature of the sample, reflecting the lowest thermal stability of the sample; T_P_ is the maximum thermal denaturation temperature (peak denaturation temperature) of the sample, reflecting the average thermal stability of the sample. The main purpose of this experiment is to study the thermodynamic properties of connective tissue in its natural state, so it was not subjected to freeze-drying dehydration. However, different water contents will affect the enthalpy value of connective tissue, and the impact on T_O_ and T_P_ is relatively small. Therefore, this study does not consider the enthalpy change of the sample. Figure 5 showed that there was only one major peak at approximately 65 °C in the thermogram, which confirmed the purity of the IMCT extracts. The IMCT thermal transition temperature was similar to that reported by Voutila et al. [38] and Latorre et al. [7].

In the control group (C-36 h) at pH values of 5.0 and 5.5 in Table 1, a slight decrease was found in both the peak temperature (T_P_) and onset temperature (T_O_) values in comparison with untreated IMCT because of the Schiff base hydrolysis, unstable collagen cross-link (aldimine bonds) breakdown, a narrower hydrated layer in the presence of Ca^2+^, and charged interactions [6,39]. This small decrease may be due to various factors: the hydrolysis of Schiff base in acidic environments, the cleavage of immature crosslinks (aldehyde imine bonds), the weakening of the hydration layer in the presence of Ca^2+^, and charge interactions between ions. After enzyme treatment, the decrease in T_O_ and T_P_ was mainly due to the hydrolysis of collagen end peptides and core proteoglycans, in addition to solvent effects. The removal of terminal peptides will inevitably be accompanied by the dissociation of mature cross-links, and the degradation of core protein glycans will also be accompanied by the dissociation of glycosaminoglycans, promoting a stable reduction in the thermal denaturation of connective tissue.

The incubation time had a significant effect (*p* < 0.05) on the T_O_ at pH 5.0 (Table 1). The T_O_ value decreased from 60.63 °C to 41.41 °C over the incubation time. Although a decrease in T_O_ was also observed at pH 5.5, it showed no significant change, except in the 36 h treatment group. With respect to the T_P_, the value gradually decreased at pH 5.0 and reached approximately 43.79 °C after 36 h of incubation. At pH 5.5, the T_P_ exhibited no significant change (*p* > 0.05).

The changes in thermal transition temperatures were likely attributable to the cleaving of telopeptides and the degradations of DCN in the extracellular matrix (ECM), as supported by the results of the collagen CNBr peptides and the DCN blots described in Section 3.2 and Section 3.3 and by the points provided by Garnero et al. [40] and Kirschke et al. [41]. Proteoglycan, as the most important proteoglycan in striated muscles, can play a guiding role in the formation of collagen fibers and serve as a spacer embedded in the collagen fiber structure. The degradation of core proteoglycans means that their modification effect on collagen fibers is weakened, and at the same time, the distributed glycosaminoglycan loses their attachment sites, resulting in a decrease in the interconnecting “bridges” between collagen fibers. This may be the reason why collagen fibers are dispersed into fine filaments. In addition, one cross-linking site was found in the telopeptide region. The cleavage of the telopeptides indicated that there were not enough cross-links, such as lysyl pyridinoline (LP) and hydroxylysyl pyridinoline (HP), to join three collagen molecules, which was also a probable reason for the decreased thermal transition temperature [40].

## 4. Conclusions

An SDS-PAGE analysis of CNBr-digested peptides showed that the degradation of collagen by purified cathepsin L from bovine pancreas could occur at the terminal non-helical peptides rather than at the triple helix region. The DCN was clearly degraded at pH 5.0 by purified cathepsin L. A specific cleavage in the collagen and the hydrolysis of DCN commonly resulted in a decrease in the thermal transition temperature (T_P_ and T_O_) of IMCT when it was incubated with purified cathepsin L. In addition, the decrease in the T_P_ and T_O_ of IMCT at pH 5.0 was more sensitive than that at pH 5.5~6.5. In this work, the influence of cathepsin L on IMCT was investigated. Moreover, the effects of other cathepsins should be considered in the future. This study explored the degradation effect of purified cathepsin L on connective tissue, improved the theory of post-mortem tenderization, and provided theoretical support for the development of new tenderizers.

## Figures and Tables

**Figure 1 foods-12-03517-f001:**
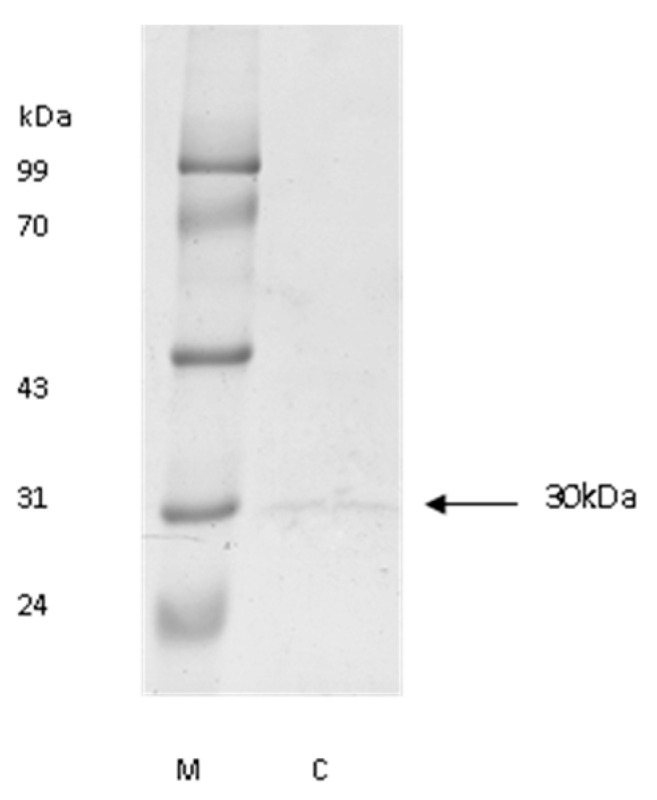
The SDS-PAGE pattern of purified cathepsin L. M, standard protein; C, the final purified cathepsin L.

**Figure 2 foods-12-03517-f002:**
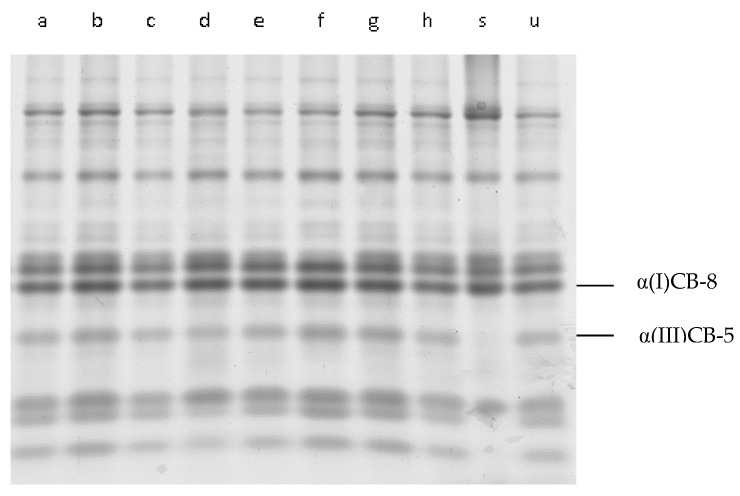
The SDS-PAGE of CNBr peptides from residual insoluble materials after the cathepsin L treatment of IMCT under different pHs during incubation (12 h). Lane (**a**) incubation pH of 5.0 without cathepsin L, lane (**b**) incubation pH of 5.5 without cathepsin L, lane (**c**) incubation pH of 6.0 without cathepsin L, lane (**d**) incubation pH of 6.5 without cathepsin L, lane (**e**) incubation pH of 5.0 with cathepsin L, lane (**f**) incubation pH of 5.5 with cathepsin L, lane (**g**) incubation pH of 6.0 with cathepsin L, lane (**h**) incubation pH of 6.5 with cathepsin L, lane (**s**) collagen type I from bovine Achilles tendon CNBr-digest standard, and lane (**u**) untreated IMCT CNBr-digested peptides.

**Figure 3 foods-12-03517-f003:**
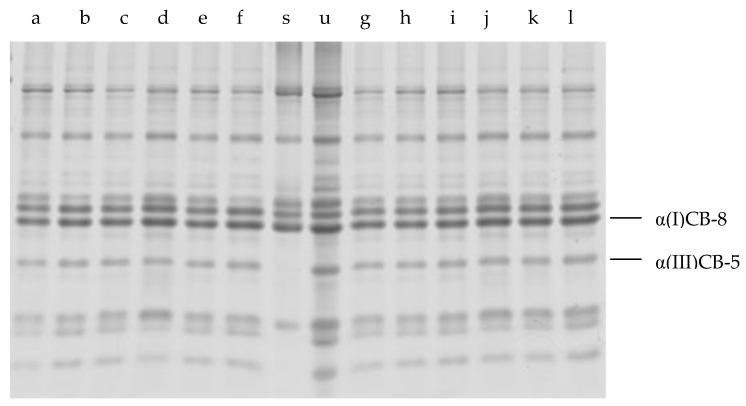
The SDS-PAGE of the CNBr peptides of residual insoluble materials after cathepsin L treatment of IMCT at different incubation times. Lane (**a**) incubation time of 12 h without cathepsin L (pH 5.0), lane (**b**) incubation time of 24 h without cathepsin L (pH 5.0), (**c**) incubation time of 36 h without cathepsin L (pH 5.0), lane (**d**) incubation time of 12 h with cathepsin L (pH 5.0), lane (**e**) incubation time of 24 h with cathepsin L (pH 5.0), lane (**f**) incubation time of 36 h with cathepsin L (pH 5.0), lane (**s**) collagen type I from bovine Achilles tendon CNBr-digested standard, lane (**u**) untreated IMCT CNBr-digested peptides, lane (**g**) incubation time of 12 h without cathepsin L (pH 5.5), lane (**h**) incubation time of 24 h without cathepsin L (pH 5.5), lane (**i**) incubation time of 36 h without cathepsin L (pH 5.5), lane (**j**) incubation time of 12 h with cathepsin L (pH 5.5), lane (**k**) incubation time of 24 h with cathepsin L (pH 5.5), and lane (**l**) incubation time of 36 h with cathepsin L (pH 5.5).

**Figure 4 foods-12-03517-f004:**
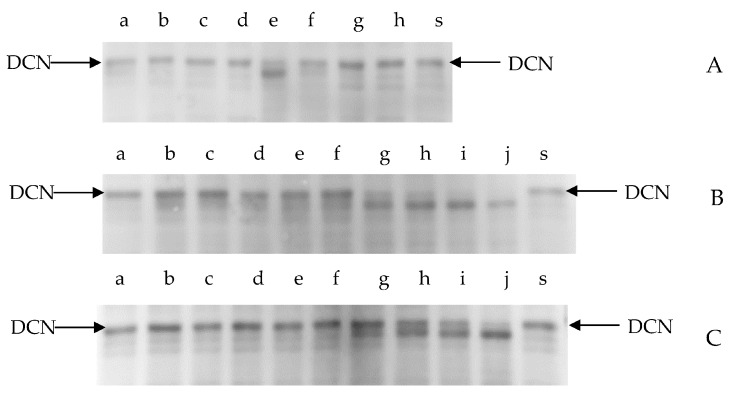
The effects of cathepsins on DCN as detected by immunoblots. (**A**) The degradation of DCN at different pHs during incubation (8 h). Lanes a, b, c, and d: incubation pH values of 5.0, 5.5, 6.0, and 6.5 without cathepsin L; lanes e, f, g, and h: incubation pH values of 5.0, 5.5, 6.0, and 6.5 with cathepsin L; and lane s: initial DCN without cathepsin L. (**B**) The degradation of DCN at different incubation times (pH = 5.0): lanes a, b, c, d, and e: incubation times of 4, 6, 8, 12, and 24 h without cathepsin L; lanes f, g, h, i, and j: incubation times of 4, 6, 8, 12, and 24 h with cathepsin L; and lane s: initial DCN without cathepsin L. (**C**) The degradation of DCN for different incubation times (pH = 5.5): lanes a, b, c, d, and e: incubation times of 4, 6, 8, 12, and 24 h without cathepsin L; lanes f, g, h, i, and j: incubation times of 4, 6, 8, 12, and 24 h with cathepsin L; and lane s: initial DCN without cathepsin L.

**Figure 5 foods-12-03517-f005:**
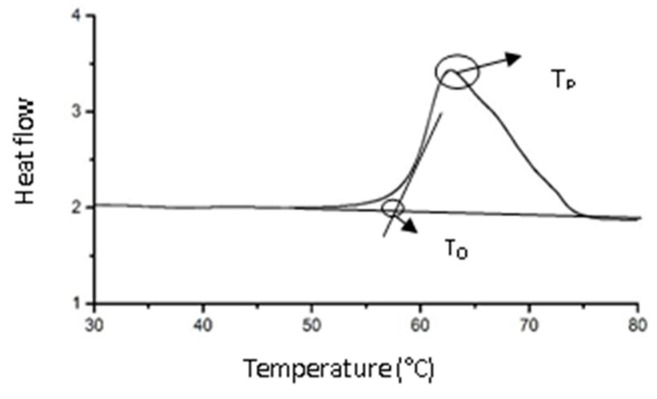
The determination of the onset and peak temperature (T_O_ and T_P_) of thermal transitions in intramuscular connective tissues.

**Table 1 foods-12-03517-t001:** The thermal transition temperature of cathepsin L-treated IMCT.

	Time			Untreated		Time			Untreated
	C-36 h	24 h	36 h			C-36 h	24 h	36 h	
T_O_-5.0 (°C)	57.87 ± 1.09 ^aX^	54.90 ± 0.82 ^bX^	41.41 ± 0.60 ^cX^	60.63 ± 0.72 ^d^	T_P_-5.0 (°C)	62.04 ± 1.02 ^aX^	60.76 ± 0.70 ^aX^	43.79 ± 0.66 ^bX^	65.38 ± 1.65 ^c^
T_O_-5.5 (°C)	59.73 ± 0.50 ^aX^	59.24 ± 0.93 ^aY^	56.57 ± 0.71 ^bY^	60.63 ± 0.72 ^a^	T_P_-5.5 (°C)	65.11 ± 0.30 ^aX^	64.80 ± 1.03 ^aY^	60.79 ± 1.01 ^aY^	65.38 ± 1.65 ^a^

c-36 h: control group treated without cathepsin over 36 h; T_O_-5.0: number represents pH value; Each value is expressed as the mean ± s.d. *n* = 3. Different letters in the same row indicate significant differences (*p* < 0.05); different capital letters in the same column indicate significant differences (*p* < 0.05).

## Data Availability

All available data are contained within the article.

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
