# Peer review of "The Degradation of Intramuscular Connective Tissue In Vitro with Purified Cathepsin L from Bovine Pancreas"

_foods, 2023, doi:10.3390/foods12183517_

Round 1

Reviewer 1 Report

Foods-2605264 Degradation of Intramuscular Connective Tissue in vitro by Purified Cathepsin L from Bovine Pancreas

I have read this manuscript and find that contains important information for future practical use in lowering bovine meat texture. I found only minor mistakes, see following list of comments.

-          Line 155 There is wrong Figure number 4. Correct number is 5.

-          Line 252 There are given lane codes “e, g, c and d”. I think that lane codes should be “e, f, g and h”.

-          Table 1 contains temperature codes To-5.0 (°C) etc. There should be added remark that number represents pH value.

-          Line 284 Sentence declares that lower pH is more convenient for reaction of collagen with Cathepsin L. I have to declare that good bovine meat can be got only when animal is not killed in stress conditions. Otherwise meat comes to dark and pH increases during ripening instead pH lowering. Subsequently, the meat spoils quickly.

-          I hope that this study can continue by experiments “in-vivo” including measuring texture parameters as true evidence of meat quality increase using Cathepsin L as the good tool.

Main reason of the study was to demonstrate the collagen and decorin proteins degradation at intramuscular connective tissue by active purified cathepsin L enzyme.

This method of bovine meat treatment would be welcomed. It enables to lower fresh meat texture. I have not done broad search in existing literature but it seems me that it was not searched in this details yet.

Paper is well written and easy to read. It is well structured and all methods used well described. Conclusions well summarize received results. And they did address the main question posed.

Reviewer 2 Report

MANUSCRIPT: 2605264

TITLE: Degradation of Intramuscular Connective Tissue in vitro by Purified Cathepsin L from Bovine Pancreas

The manuscript 2605264 “Degradation of Intramuscular Connective Tissue in vitro by Purified Cathepsin L from Bovine Pancreas presents an interesting study in order to demonstrate the degradation of collagen and decorin in intramuscular connective tissue by uprified Cathepsin L.

This work is well structured, well planned and the research is competently carried out, the methodology was quite adequate to the research.

The literature is well cited and most of the papers cited (24.4 %) date back to the last five years and it was made a statistical analysis.

However, the manuscript needs to be improved in some aspects, so the following points for review should be considered:

11. Abstract – Please reformulate the abstract. The abstract must be a summary of the entire manuscript and must include the main idea of the introduction, identify the main materials and methods, as well as the main results and conclusions.

Materials and Methods

S2. Section 2. Materials and Methods – Line 76, centrifuge used. Please carefully review this section and in all subsections consider add the on each instrument used, model, producer and its location (Instrument model, Producer, City, State Abbr., Country). Proceed in the same way for all instruments used.

 3. Lines 78 and 84 – Please it is necessary to identify after centrifuged the conditions of time, temperature and centrifugal force used.

  4. Please, to allow the replication of the study, it is necessary to describe in detail the methods for each parameter determined, including in all of them the complete description of the methodologies including sample quantities and presentation of equations for determining the parameters.

  5.  Section 2.2 Preparation of Purified Cathepsin L . Please describe in detail the method in this section namely how chromatography is performed by DEAE Sephacel, Sephacryl S-100 87 HR, SP Sepharose FF and con A Sepharose affinity chromatography (lines 87-88) and how the optimum pH is determined (line 93).

  6. Section 2.3 Incubation of IMCT with Purified Cathepsin L, CNBr Digestion and SDS-PAGE. Please describe in detail the method in this section namely how the sample is prepared and the conditions under which SDS-PAGE is carried out to separate the peptides as well as indicating and identifying the composition of the molecular weight markers.

  7. Section 2.4 Western Blot Analysis of the Degradation of Decorin (DCN) Incubated with Cathepsin L in vitro. Please describe in detail the method in this section namely how the protein concentration is estimated (line119).

   8. Statistical analysis section – Please indicate the (n) number of replicates in each experimental parameter determined and carry out a comparative statistical analysis in order to determine whether there are or not there are statistically significant differences between the results obtained in all situations where it is possible to carry out this analysis.

Results and discussion

9. Figure 1 caption – Please, it is recommended indicate what are the lanes identified by M and C and present for molecular weight markers their composition, that is, identify for each of the markers from 24 to 99 kDa which protein or polypeptide corresponds to the respective band.

10. Table 1 - Please in the footnote, in addition to the mean ± SD, indicate the number of repetitions of the experiments for each given parameter (n=?).

Other points

11.  Line 31 - Please - Latin expressions should be written in italics, I would recommend writing the expression "et al" throughout the manuscript in italics.

12.  Line 35 (To and Tp) – Please write it in full when writing an abbreviation or symbol for the first time.

13. Line 65 - Please - Latin expressions should be written in italics, I would recommend writing the expression " Longissimus lumborum " throughout the manuscript in italics.

14.  Line 68 – Please change “4ËšC” by “4 ËšC”. Please review the manuscript and give space between the centigrade degree symbol (ËšC) and the numerical value.

15.  Line 72 – Please when writing an abbreviation (PMSF) for the first time you must write it in full.

16.  Line 91 – Please when writing an abbreviation (AMC) for the first time you must write it in full.

17.  Line 133 – Please when writing an abbreviation (PVDF) for the first time you must write it in full.

18.  Line 135 – Please when writing an abbreviation (TBST) for the first time you must write it in full.

19.  Line 270 – Please change “Ca2+” by “Ca2+”.
